# Autism Spectrum as an Etiologic Systemic Disorder: A Protocol for an Umbrella Review

**DOI:** 10.3390/healthcare10112200

**Published:** 2022-11-02

**Authors:** Lara Teixeira Lopes, Jorge Magalhães Rodrigues, Celeste Baccarin, Kevin Oliveira, Manuela Abreu, Victor Ribeiro, Zélia Caçador Anastácio, Jorge Pereira Machado

**Affiliations:** 1Institute of Biomedical Sciences, University of Porto, 4099-030 Porto, Portugal; 2Center of BioSciences in Integrative Health, 4200-355 Porto, Portugal; 3Research Center on Child Studies, Institute of Education, University of Minho, 4804-533 Braga, Portugal

**Keywords:** autism, ASD, umbrella review, causes, aetiology, etiology, systemic disorder, pathogenesis, health policy, integrated care services

## Abstract

Autism spectrum disorder (ASD) is the most common neurodevelopmental disorder with a huge prevalence increasing every year (1/44 children). Still diagnosed as a mental disorder, the last 10 years of research found possible causes, risks, genetics, environmental triggers, epigenetics, metabolic, immunological, and neurophysiological unbalances as relevant aetiology. Umbrella methodology is the highest level of scientific evidence, designed to support clinical and political decisions. A literature search for autism aetiology, pathophysiology, or causes, conducted in the last 10 years, at PubMed, Embase, Cochrane, Scopus, and the Web of Science, resulted in six umbrella reviews. Nevertheless, only one quantitative analysis reported risk factors and biomarkers but excluded genetics, experiments on animal models, and post-mortem studies. We grouped ASD’s multi-factorial causes and risks into five etiological categories: genetic, epigenetic, organic, psychogenic, and environmental. Findings suggest that autism might be evaluated as a systemic disorder instead of only through the lens of mental and behavioural. The overview implications of included studies will be qualitatively analysed under ROBIS and GRADE tools. This umbrella review can provide a rational basis for a new urgent health policy to develop better and adequate integrated care services for ASD. The methodological protocol has the register CRD42022348586 at PROSPERO.

## 1. Introduction

### 1.1. Autism Spectrum Disorder

Autism spectrum disorder (ASD) is the most common neurodevelopmental disorder, affecting 1/44 children in 2018 [1,2]. It is diagnosed as a mental disease, but research from the last 10 years reports more and more biological causes. Autism costs an estimated USD 3.2 million [3] in a lifetime per child, according to the Centers for Disease Control and Prevention in the USA [1].

ASD is a behavioural condition with mental implications, according to the criteria defined in the *Diagnostic and Statistical Manual of Mental Disorders—Fifth Edition* (DSM-5; APA 2013). It is diagnosed by impairments in social communication or the presence of restricted or repetitive behaviours/interests, or both. Diagnosis is made according to classification systems. Tests to diagnose ASD have been developed using a parent or caregiver interview, child observation, or a combination of both, being the so-called validated psychometric scales [4]. Despite that, research advancements confirm that ASD aetiology is multifactorial.

### 1.2. Genetic Aetiology

A systematic review of sex differentiation found interesting links between risk genes and the potential role of estrogens as modulators of biological pathways in ASD, and highlights relevant molecular and cellular pathways downstream of estrogen signaling as potential avenues for further investigation [5].

Moreover, environmental and genetic risk factors, such as maternal exposure to pesticides [6,7] and women aged above 35 years old [8], may intensify vulnerability to oxidative stress [8,9,10,11].

### 1.3. Epigenetic Aetiology

So, the development of ASD pathogenesis and clinical symptoms are probably connected to increased oxidative stress.

Oxidative stress at a membrane level, the antioxidant products involved in the defense system against reactive oxygen species (ROS), the detoxifying agents (like glutathione), and the products of lipid peroxidation, are some of the studied epigenetic possible causes. Other studies indicate some alterations in the antioxidant enzymes activities may be involved, such as superoxide dismutase, glutathione peroxidase, and catalase [5,12,13]. It is also suggested that there can be alterations in homocysteine/methionine metabolism and glutathione levels, mitochondrial and immune dysfunction, excitotoxicity, as well as increased inflammation [8,9].

Furthermore, increased oxidative stress seems to be linked to the development of ASD in terms of both pathogenesis and clinical symptoms. On the other hand, antioxidant supplementation, or ways to improve the altered metabolite levels in the interconnected transmethylation and transsulfuration pathways, has been associated with decreased autistic behaviours and severity [12].

Darko Sarovic recently proposed a “unifying theory for Autism” focusing on the interaction between exogenous and immunological burdens affecting ASD people, and concluded by applying the framework of a pathogenic triad to the scientific literature. It proposes a deconstruction of autism into three contributing features: (1) an autistic personality dimension; (2) cognitive compensation; and (3) neuropathological risk factors. It delineates how they interact to cause a maladaptive behavioural phenotype [14].

### 1.4. Organic Aetiology

Research concerning biochemical biomarkers [15,16] reported immune system alterations [17], mitochondrial function [18,19], and, oxidative stress pathways [12], as common issues in ASD. In the last years, special attention to gut dysbiosis, microbiota imbalances/abnormal pathogenic bacterial proliferation, and interaction with brain disorders, has been increasing. Indeed, some authors are so convinced about the huge influence of the “gut-brain axis” over emotions, behaviour, biochemical, and neurophysiological regulation, that they name the gut the “second brain” [20,21]. The U.S.A. medical emergency services often diagnose a bacterial predisposed psychiatric disease known as PANDAS (Pediatric Autoimmune Neuropsychiatric Disorders Associated with Streptococcal Infections) characterized mostly by sudden behaviour alterations/OCD (obsessive-compulsive and/or tic disorder) after streptococcal (strep) infection, such as strep throat or scarlet fever [6,22,23]. This indicates a strong interdependent two-way link between gut and behaviour, thus relevant enough to inspire deeper research about the path-ways involved.

These changes in the mechanisms of absorption, or the permeability of the intestinal epithelial mucosa, can lead to an inflow of toxic or foreign organic or inorganic substances causing changes in immunological or metabolic defenses. That can ultimately affect the nerve pathways at the level of the large gastrointestinal nervous complex, or extend directly or indirectly to the central nervous system [24,25].

### 1.5. Psychogenic Aetiology

In terms of brain electrophysiology, autism has been characterized by atypical task-related brain activation and functional connections, coinciding with deficits in socio-communicative abilities [26] and several neurotransmitter systems [27,28].

The endocannabinoid system is a hot topic on mental disorders pathways; an interesting review indicates overlapping bio-behavioural aberrancies between autism and schizophrenia. In addition, common cannabinoid-based pharmacological strategies have been identified, exerting epigenetic effects across genes and controlling neural mechanisms [29].

### 1.6. Environmental Aetiology

People with ASD are exposed to the negative effects of chemical and environmental risks, such as aluminium from food packaging and cadmium from air pollution [7]. Since several studies report that ASD people struggle to eliminate toxins from the body, there may be an increased probability of a positive effect from taking antioxidant supplements [24,25].

### 1.7. Future Steps for Integrated Care Services

Searching the literature for umbrella reviews about possible causes, pathological pathways, and/or aetiology of ASD, the authors obtained six papers related to umbrella methodology on this topic. Nevertheless, only one is aetiology concerned. That review quantitively analyzed biomarkers and risk factors but excluded genetics and experiments in animal models, as well as in vitro and postmortem studies. As the authors decided on these exclusions, a big part of recent research contributing to the understanding of ASD pathways was not reported. Moreover, several possible biomarkers are not described because studies with less than 1000 participants were considered insignificant. The main conclusion of this study was that pre-maternal obesity, metabolic, circulatory, and psychiatric disorders (such as depression and anxiety or the pre-pregnant use of antidepressant drugs, such as serotonin recaption inhibitors, as a confounding factor) are correlated with more ASD newborns [30].

Despite all the recent holistic frameworks, ASD is still being clinically addressed as a mental disorder, leading to treatment mostly based on cognitive behaviour-based therapies. These seem to be effective, however, only when given many hours (e.g., 40 h/week for ABA Therapy) [31].

Pervasive neurodevelopmental diseases dramatically increased over the last decades and most of the available therapies are, as mentioned above, tailored to modulate behaviour, support emotional regulation, and change environmental contexts to be more autistic friendly. The extreme relevance of all these validated therapies is undiscussable; however, they work mostly outside-oriented. Many international evidence-based clinical practice guideline programs have highlighted ASD as one of the top health priorities [32].

Pragmatic awareness and clinical staff education about organic/systemic aetiology need to be urgently discussed in order to define an updated and integrative therapeutic approach.

With this neurodevelopment disorder’s prevalence increasing, it is vital that we change the way of looking into it. The knowledge of risks, causes, biomarkers, and all the already discovered pathophysiology mechanisms in the last years, contributes to a great improvement in ASD aetiology understanding. Nevertheless, politicians, decision-makers, and clinicians need to have access to this information. This review intends to highlight an important summary to help them to act faster. To effectively ameliorate ASD people’s quality of life and to reduce family/caregiver stress related to the management of autism-challenging behaviours, this umbrella review evidence report aims to provide a rational basis to contribute to new health policies and better-integrated care.

## 2. Materials and Methods

This protocol was developed according to the Preferred Reporting Items for Systematic Reviews and Meta-Analyses Protocols (PRISMA-P) guidelines. Following these guidelines, the present umbrella review protocol has been registered in the International Prospective Register of Systematic Reviews with CRD42022348586. Protocol amendments will be documented in PROSPERO. The procedures, steps, team, and tasks of the umbrella review, to which this protocol refers, are shown in Figure 1.

### 2.1. Search Strategy

#### 2.1.1. Database Search

Five independent researchers will search Pubmed, Embase, Cochrane, Scopus, and the Web of Science databases for the last three years of published meta-analysis and systematic reviews. Although we will filter publishing dates to 2020, 2021, and 2022, data extracted will be exclusively from meta-analysis and systematic reviews, and the study methodologies report enough information regarding, at least, the last 10 years of findings.

The PICOT (population, issue/intervention, conditions, outputs, and timeframe) assessment of objectives was performed to assure keywords and commands are enough to retrieve significant data, as reported in Table 1.

Endnote references software will be used to export and format all the citations. Duplicate references will be automatically selected and deleted.

#### 2.1.2. Search Terms

The following keywords will be searched: (autism, ASD) AND (a*etiology, causes, association*(s), risk factors, biomarkers, pathophysiology, pathogenesis, pathological mechanism*(s), and pathway*(s)).

Filters: last three years of the publication date, only meta-analysis, and systematic reviews as methodological designs. In the PubMed database to ensure that only those two designs are searched, we will use the command ‘systematic[sb]’.

### 2.2. Eligibility Criteria

#### 2.2.1. Study Design

This study protocol is a review of reviews design using qualitative analysis. This design type is the highest level of evidence and is best known as the umbrella review [33,34].

#### 2.2.2. Inclusion/Exclusion Criteria

In this methodology, only meta-analysis and systematic reviews are accepted for inclusion.

To answer our objectives, reviews addressing the possible causes and aetiology of autism spectrum disorders are eligible. ASD people, genetics, experiments in animal models, postmortem, as well as in vitro studies, were to be included.

Other review types (not systematic) and isolated clinical trials will be excluded. Systematic reviews or meta-analyses that do not bring any clear etiological insight related to ASD, or are not published in English full text, will be also excluded.

#### 2.2.3. Study Selection

The study selection is to be handled in EndNote and in the intelligent systematic review software program named Rayyan. The first author will perform the initial search and the removal of duplicates. In all the identified publications, five independent researchers, in blinded mode, will carry out a screening of titles and abstracts for inclusion and exclusion criteria. If the eligibility of the publication is unsure, the publication must be identified with a maybe tag on the Rayyan software and should be enrolled in the next selection step, to be decided by all the researchers in the final selection meeting. All papers identified as potentially relevant by at least one researcher must be retrieved in full text and assessed for eligibility by the four other independent researchers. In case of disagreement, the five researchers discuss it and, if necessary, a senior invited researcher might be included in the final decision.

Finally, the reference lists of the included publications are manually screened to identify manuscripts which are eligible, according to the inclusion criteria. If both an original version and an updated version of a systematic review are identified, both papers should be included and discussed in the umbrella review. In our review of reviews, a PRISMA flow diagram was to show the selection flow and the number of included publications.

#### 2.2.4. Data Extraction

Citations saved at EndNote software will be exported in RIS format and uploaded to a Rayyan shared folder. The Rayyan software was chosen to decide under a blinded methodology between the five researchers on the inclusion and exclusion of articles after title, keywords, and abstract reading individually.

Disagreements are to be solved at a meeting between all researchers.

To summarize the process, a PRISMA flowchart presenting the sequence of manuscripts selected at each screening level will be included according to the procedures of several authors, such as Catallo et al., 2022 [35].

After the selection of articles according to eligibility criteria, they will be randomly distributed among the five researchers for full reading and qualitative analysis.

A shared excel table, exported from *Rayyan* and automatically screening including: (a) citation, (b) title, (c) abstract, and (d) classification (1 to 5 stars), according to a significance level of keywords of inclusion, must be completed per article with a summary of evidence to extract.

#### 2.2.5. Assessment of Methodological Quality

To avoid quality assessment bias, the ROBIS tool for umbrella reviews will be used. A checklist of ROBIS questions should be provided to the five researchers for bias risk and the quality control of each article included in the final report [36].

Three researchers were to screen and judge the quality of each selected review or meta-analysis, according to GRADE guidelines using the GRADEpro GDT software from McMaster University and Evidence Prime (2022) to generate, record, and publish the following tables: evidence profile; a summary of findings; evidence to decision framework and an interactive summary of findings. This is a grading system to reduce bias by methodological qualifying findings as very low, low, moderate, or high evidence [37].

#### 2.2.6. Results

From the five databases searched, 9911 registers were obtained. After EndNote duplicates removal, 663 records remain. Applying Rayyan’s ineligible automation tool, 442 references were excluded and 241 were included at level 1 screening.

Research results will be presented in a complete PRISMA flowchart. To date, two, of the five researchers completed the first level of blinded screening at Rayyan software. Researcher 1 included 189 articles but researcher 2 selected only 159 papers. Those previous results are reported in Figure 2, a PRISMA flowchart.

Tables for synthesis presenting each of the five etiologic categories will be carried out according to the GRADE summary of findings guidelines, like the example in Table 2. A biorender picture must be designed to summarize pathways and physiological mechanisms implicated in ASD development. If moderate to high GRADEs of evidence are founded, evidence to decision framework tables was to be also included [37].

Figure 3 schematic presents our proposal for the arrangement of Results in five etiologic categories.

## 3. Discussion

The literature reports several advantages regarding this type of protocol publishing: (a) increased clarity surrounding systematic review conduct and reporting [33,38]; (b) it helps to reduce the bias of “positive” reviews selection; and (c) avoids the duplication of the research team’s efforts by making protocol publicly available, while conducting with more time the umbrella review.

In a study where five researchers independently search for the same criteria, it is highly important to precisely define, step by step, objectives, methodology, search strategy, filters, and resources. For that purpose, a protocol is a useful guideline. Moreover, other researchers might benefit from accessing this kind of publication to help structure methodological reproducibility for umbrella reviews in mental or psychiatric conditions [39,40]. Even though the definition of research questions, population, outputs, and timeframe, are well described in the PICOT table, in previous protocols, researchers report a moderate level of disagreement about which articles to include or exclude [35]. This suggests that reliability across researchers should be improved after further meetings to discuss disagreements under this protocol application

Results have to be discussed with the methodological quality assessment guided by the ROBIS and GRADE tools [36,37].

The umbrella review should provide a high-level field synthesis and discuss the results from systematic reviews and meta-analyses for each possible risk or cause of autism spectrum disorder. Pathways and pathological mechanisms identified in the included systematic reviews and meta-analyses must be equally described and addressed.

We will also focus on several suggestions of therapeutic and treatment options to address the aetiology discussed in the systematic reviews and meta-analysis.

The authors hope that this umbrella review might contribute to helping decision-makers, politicians, and clinicians, to address the urgent restructuration of ASD diagnosis with consequently more effective tailor-made care services.

## 4. Conclusions

ASD is a complex multifactorial disorder. In the last decade, there have been huge findings concerning genetic, epigenetic, organic/immunological deficits, and chemical/environmental burdens as evident causes of neuro-inflammation arouse.

A framework of evidence is urgently needed as a clinical and political decision-making tool.

## Figures and Tables

**Figure 1 healthcare-10-02200-f001:**
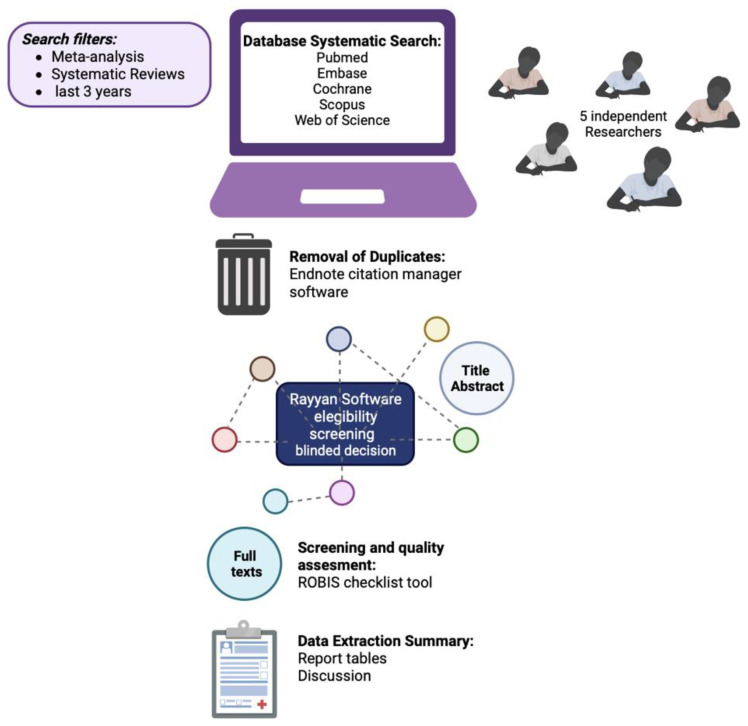
Picture chart of the protocol for the umbrella review steps. Created at 2022 Biorender, online software, Toronto, Canada.

**Figure 2 healthcare-10-02200-f002:**
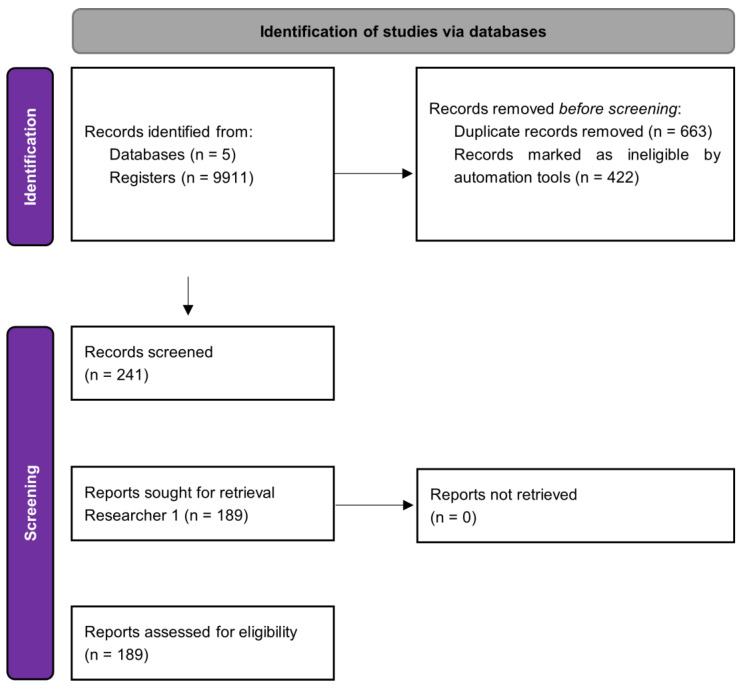
PRISMA flowchart for up-to-date results.

**Figure 3 healthcare-10-02200-f003:**
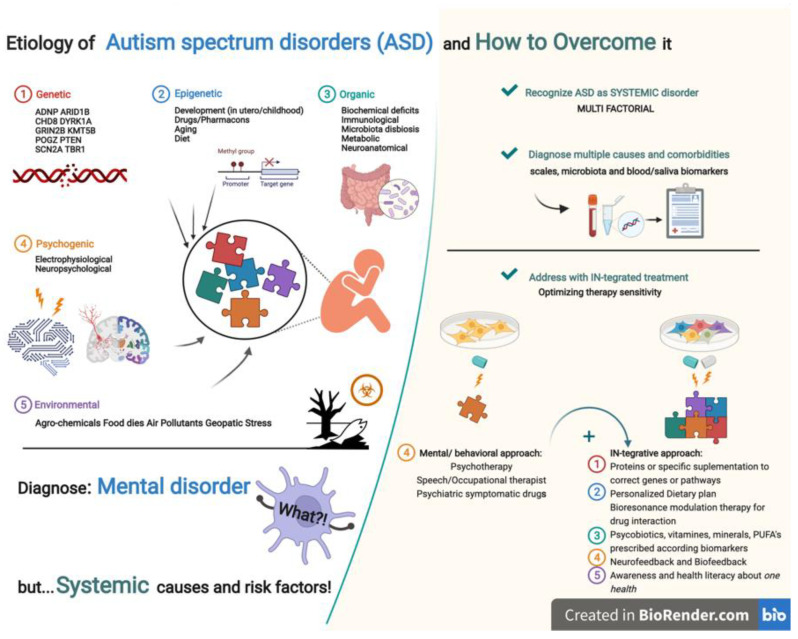
ASD aetiology categories and perspectives about IN-tegrative therapeutic approach need. steps. Created at 2022 Biorender, online software, Toronto, Canada.

**Table 1 healthcare-10-02200-t001:** PICOT approach.

PICOT Research Question and Searching Keywords Analysis for ASD Aetiology as a Systemic Disorder	
Population	Autistic: Children Adults Animal models Postmortem people In vitro: Cell and bacteria cultures
Issue addressed	Aetiology Etiology Causes Risks Biomarkers Pathological mechanisms Pathogenesis Pathophysiology Factors
Conditions	n/a
Outputs	Nº of eligible studies for each etiological category: 1. Genetic 2. Organic 3. Epigenetic 4. Psychogenic 5. Environmental Qualitative GRADE/ROBIS tables: (a) Evidence profile (b) Summary of findings (c) Evidence to decision framework (d) Interactive summary of findings (available only at GRADEpro GTD website)
Timeframe	Meta-analysis and Systematic Reviews published between 2020–2022

**Table 2 healthcare-10-02200-t002:** Example of Summary of Findings (using GRADE). Population: ASD Issue: aetiology Outputs: Nº and rating of studies in each category T: last three years published papers (2020, 2021, 2022).

Outcomes (Causes)	Population	Nº of Studies	Nº ofParticipants	Quality of the EvidenceGRADE	Comments	Results/Findings
Genetic	Animal models:Rats, fish, and frogs	389	10,890	high 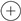 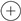 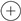 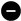		Past 10 years of remarkable progress in ASD risk gene discoveryPathway mechanisms involving these genes identificationRegulation of gene expression
Epigenetic	Animal models:rodents	…	…	moderate 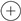 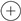 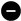 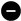		Estrogens in vivo animal models modulate high levels of testosterone and ASD behaviour decreases the correlation
Organic	Human beings:Children 3–9Adults > 19Postmortem studies (cadaverines).	…	…	Low 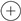 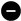 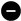 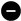		Insights into biological processes disrupted in ASD
Psychogenic	…	…	…	moderate 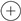 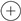 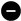 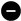		SynaptogenesisExcitatory-inhibitory imbalance
Environmental	…	…	…			

Important Note: data on this table is merely an example, not real or confirmed findings.

## Data Availability

Lara Lopes, Jorge Rodrigues, Celeste Baccarin, Victor Ribeiro, Manuela Abreu, Zélia Caçador, Jorge Machado. Autism Spectrum Disorder (ASD) aetiology: an umbrella review. PROSPERO 2022 CRD42022348586 Available from: https://www.crd.york.ac.uk/prospero/display_record.php?ID=CRD42022348586 (accessed on 30 October 2022).

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
