# Peer review of "Autism Spectrum as an Etiologic Systemic Disorder: A Protocol for an Umbrella Review"

_healthcare, 2022, doi:10.3390/healthcare10112200_

Round 1
Reviewer 1 Report
The aim of Lopes et al's study is to describe a systematic review of ASD possible causes and risks. Despite the paper's relevance, there are many deficiencies. It is a very short and weak review that is not systematic, very hard to understand what the authors want to present to support their aim, and the results are not well defined.
Comments:
1)The following types of publications are available through the Healthcare MPDI:
- Articles: Original research manuscripts. The journal considers all original research manuscripts provided that the work reports scientifically sound experiments and provides a substantial amount of new information. Authors should not unnecessarily divide their work into several related manuscripts, although short Communications of preliminary, but significant, results will be considered. The quality and impact of the study will be considered during peer review.
- Reviews: These provide concise and precise updates on the latest progress made in a given area of research. Systematic reviews should follow the PRISMA guidelines.
- Case reports: Case reports present detailed information on the symptoms, signs, diagnosis, treatment (including all types of interventions), and outcomes of an individual patient. Case reports usually describe new or uncommon conditions that serve to enhance medical care or highlight diagnostic approaches.
2)The "Protocol for a Umbrella Review" should be changed to only review or systematic review.
3)Two whole paragraphs were reused in the introduction; Lanes 57-70 matched completely with a previous publication in https://link.springer.com/chapter/10.1007/978-3-030-30402-7_7
In paraphrasing and/or summarizing others' work, we must ensure that we are reproducing the exact meaning of the other author's ideas, not copying and pasting.
4)It is not necessary to write objectives in a section, but simply mention the main goal and highlight the main conclusions at the end of the introduction.
- Manuscript sections: Introduction, Materials and Methods, Results, Discussion, Conclusions (optional).
- Introduction: The introduction should briefly place the study in a broad context and highlight why it is important. It should define the purpose of the work and its significance, including specific hypotheses being tested. The current state of the research field should be reviewed carefully and key publications cited. Please highlight controversial and diverging hypotheses when necessary. Finally, briefly mention the main aim of the work and highlight the main conclusions. Keep the introduction comprehensible to scientists working outside the topic of the paper.
5)Remove this paragraph, no value added: "This protocol is a guideline for a future Umbrella review that we aim to publish in a near future, if possible, until the end of 2022, on an open access Journal to guarantee free access to this knowledge."
6)Result: this section is quite disappointing, there are no results, just three sentences referring to figures and tables without describing the actual results. There should be no bias or interpretation in the results section, and the findings should be arranged logically.
7)In science manuscripts, the information is presented in the past tense not future tense. Editing the English language is required throughout the manuscript due to too many mistakes.
Author Response
Dear Reviewer nº 1,
We acknowledge you for your review report and we have considered your comments to improve our manuscript. We further answer below in the end of each of your relevant comments.
Comments:
1)The following types of publications are available through the Healthcare MPDI:
- Articles: Original research manuscripts. The journal considers all original research manuscripts provided that the work reports scientifically sound experiments and provides a substantial amount of new information. Authors should not unnecessarily divide their work into several related manuscripts, although short Communications of preliminary, but significant, results will be considered. The quality and impact of the study will be considered during peer review.
- Reviews: These provide concise and precise updates on the latest progress made in a given area of research. Systematic reviews should follow the PRISMA guidelines.
- Case reports: Case reports present detailed information on the symptoms, signs, diagnosis, treatment (including all types of interventions), and outcomes of an individual patient. Case reports usually describe new or uncommon conditions that serve to enhance medical care or highlight diagnostic approaches.
2)The "Protocol for a Umbrella Review" should be changed to only review or systematic review.
Points 1 and 2:
Healthcare Guidelines for Authors refers the following types of manuscripts as acceptable:
https://www.mdpi.com/about/article_types
- Article
- ...
- Protocol (Protocols provide a detailed step-by-step description of a method. They should be proven to be robust and reproducible and should accompany a previously published article that uses this method. Any materials and equipment used should be explicitly listed. Conditions, quantities, concentrations, etc., should be given. Critical timepoints and steps, as well as warnings, should be emphasized in the text. The structure should include an Abstract, Keywords, Introduction, Experimental Design, Materials and Equipment, Detailed Procedure, and Expected Results, with a suggested minimum word count of 4000 words.)
- Registered report
- ...
In fact, we also found a scoping review protocol recently published at Healthcare mdpi Journal: https://doi.org/10.3390/healthcare10101847
So authors decide to maintain the protocol methodology.
3)Two whole paragraphs were reused in the introduction; Lanes 57-70 matched completely with a previous publication in https://link.springer.com/chapter/10.1007/978-3-030-30402-7_7
In paraphrasing and/or summarizing others' work, we must ensure that we are reproducing the exact meaning of the other author's ideas, not copying and pasting.
Point 3: authors rewrite this paragraph according to your suggestions.
Authors rewrite these ideas.
4)It is not necessary to write objectives in a section, but simply mention the main goal and highlight the main conclusions at the end of the introduction.
- Manuscript sections: Introduction, Materials and Methods, Results, Discussion, Conclusions (optional).
- Introduction: The introduction should briefly place the study in a broad context and highlight why it is important. It should define the purpose of the work and its significance, including specific hypotheses being tested. The current state of the research field should be reviewed carefully and key publications cited. Please highlight controversial and diverging hypotheses when necessary. Finally, briefly mention the main aim of the work and highlight the main conclusions. Keep the introduction comprehensible to scientists working outside the topic of the paper.
Point 4: following your suggestions, authors removed the subtitle objectives, rewrite and highlight the main conclusions.
5)Remove this paragraph, no value added: "This protocol is a guideline for a future Umbrella review that we aim to publish in a near future, if possible, until the end of 2022, on an open access Journal to guarantee free access to this knowledge."
Point 5: paragraph removed.
6)Result: this section is quite disappointing, there are no results, just three sentences referring to figures and tables without describing the actual results. There should be no bias or interpretation in the results section, and the findings should be arranged logically.
7)In science manuscripts, the information is presented in the past tense not future tense. Editing the English language is required throughout the manuscript due to too many mistakes.
Reviewer 2 Report
The protocol of Lopes et al. focuses on an always current topic of the etiology of ASD. My main concern is if it would not be better to publish an umbrella review together with a protocol instead of dividing it into two manuscripts.
Additionally, I had the following comments:
Introduction-several aspects are discussed, some of them are not well developed and described in 2-3 sentences. I would recommend revising the introduction, group the threads and avoid the repetitions (e.g. regarding the benefits of an umbrella review).
Line 60 several studies - only one study is cited
Line 65 "environmental and genetic risk factors"-please give an example of them
Line 71-72-please develop the idea
Line 74-" a recent umbrella review"-how many umbrella reviews are there in total?
Line 80-which psychiatric disorders?
Objective:
Line 115-117-"This review of reviews intends to highlight an important summary to help them 116 act faster"-the keywords are more regarding pathophysiology and biomarker than regarding policymakers
Please present the number of studies in each category
Eligibility criteria
Are animal studies included? Only in the English language?
Please add 1-2 conclusion sentence
Author Response
Dear Reviewer nº 2,
We acknowledge you for your review report and we have considered your comments to improve our manuscript. We further answer below in the end of each of your relevant comments.
Comments:
My main concern is if it would not be better to publish an umbrella review together with a protocol instead of dividing it into two manuscripts.
Point 1: Healthcare Guidelines for Authors refers the following types of manuscripts as acceptable:
https://www.mdpi.com/about/article_types
- Article
- ...
- Protocol (Protocols provide a detailed step-by-step description of a method. They should be proven to be robust and reproducible and should accompany a previously published article that uses this method. Any materials and equipment used should be explicitly listed. Conditions, quantities, concentrations, etc., should be given. Critical timepoints and steps, as well as warnings, should be emphasized in the text. The structure should include an Abstract, Keywords, Introduction, Experimental Design, Materials and Equipment, Detailed Procedure, and Expected Results, with a suggested minimum word count of 4000 words.)
- Registered report
- etc ...
In fact, we also found a scoping review protocol recently published at Healthcare mdpi Journal: https://doi.org/10.3390/healthcare10101847
So authors decide to maintain the protocol methodology as we found on literature several advantages regarding this type of protocols publishing:
- Increased clarity surrounding systematic review conduct and reporting (PLoS Medicine Editors. Best practice in systematic reviews: the importance of protocols and registration. PLoS Med. 2011 Feb;8(2):e1001009. doi: 10.1371/journal.pmed.1001009. Epub 2011 Feb 22. PMID: 21364968; PMCID: PMC3042995.)
- Help to reduce bias of "positive" reviews selection
- Avoid duplication of researcher teams efforts by making protocol public available while conducting with more time the umbrella review
Additionally, I had the following comments:
Introduction-several aspects are discussed, some of them are not well developed and described in 2-3 sentences. I would recommend revising the introduction, group the threads and avoid the repetitions (e.g. regarding the benefits of an umbrella review).
Point 2: suggestions addressed
Line 60 several studies - only one study is cited
Point 3: suggestions addressed
Line 65 "environmental and genetic risk factors"-please give an example of them
Point 4: suggestions addressed
Line 71-72-please develop the idea
Point 5: suggestions addressed
Line 74-" a recent umbrella review"-how many umbrella reviews are there in total?
Point 6: 6 umbrella reviews related to autism but only 1 addressing partial etiology
Line 80-which psychiatric disorders?
Point 7: enumeration of which psychiatric disorders added to the text (depression and anxiety or the confounding factor related to antidepressant pre-pregnancy drug taking)
Objective:
Line 115-117-"This review of reviews intends to highlight an important summary to help them 116 act faster"-the keywords are more regarding pathophysiology and biomarker than regarding policymakers
Point 8: according to your highly valuable suggestion authors included "health policy" and "integrative care services" as keywords
Please present the number of studies in each category
Point 9: nº of studies will be achieved only in the final research for the review
Eligibility criteria - Are animal studies included? Only in the English language?
Point 10: animal studies were in included as eligibly criteria and articles with no English full text as exclusion criteria.
Please add 1-2 conclusion sentence
Point 11: suggestions addressed
Round 2
Reviewer 2 Report
I would like to thank the authors for introducing changes quickly. I think that the protocol is more understandable now. I thought about how to improve the manuscript and read an example protocol provided by the authors. I would suggest introducing the following changes:
Major changes
Results
As in the following protocol https://doi.org/10.3390/healthcare10101847 l I would suggest providing the numbers of articles included and excluded after every step.
I have the feeling that the search and filtering have not been performed so that is why a result section seems to be more a plan than a real result. Maybe including a PRISMA chart in the Results section would be a solution?
Conclusion I think that they are repeating the introduction and are not providing any comments on the number of articles in various sections that can illustrate scientific intrerest.
Discussion I think that its very short. It would be better to add some more aspects, comment on results or even highlight the pros of umbrella review (as it has been well done in the answer to reviewer)
General
I still think that it would be better to write the review first and, even if the protocol would be published separately, it can contain a link to the review.
Minor comments:
line 163 its a protocol, not a review
164-165-I am not sure if this review can provide basis for a new health policy, depends on the results
232 to exclude-I think that a verb is missing here
Table 2 number of participants-ais it possible to provide it so exact for animals
Author Response
Dear reviewer,
Thank you very much for the time you spent with our manuscript and for your good suggestions in order to improve it.
We did the changes and are giving here some answers to your comments. All the changes are signalized with different colour.
Comments and Suggestions for Authors
I would like to thank the authors for introducing changes quickly. I think that the protocol is more understandable now. I thought about how to improve the manuscript and read an example protocol provided by the authors. I would suggest introducing the following changes:
We are really very grateful for your opinion and suggestions.
Major changes
Ok, we agree perfectly, it is to improve the manuscript.
Results
As in the following protocol https://doi.org/10.3390/healthcare10101847 l I would suggest providing the numbers of articles included and excluded after every step.
Very good suggestions. We read this articles and it helped us very much to complete our manuscript. In this sense we considered appropriate to cite this article.
I have the feeling that the search and filtering have not been performed so that is why a result section seems to be more a plan than a real result. Maybe including a PRISMA chart in the Results section would be a solution?
Yes! Thank you for the suggestion. It is done now.
Conclusion I think that they are repeating the introduction and are not providing any comments on the number of articles in various sections that can illustrate scientific interest.
Discussion I think that its very short. It would be better to add some more aspects, comment on results or even highlight the pros of umbrella review (as it has been well done in the answer to reviewer)
Great suggestion. We add it to discussion and also added 2 more citations reinforcing the advantages of protocol.
General
I still think that it would be better to write the review first and, even if the protocol would be published separately, it can contain a link to the review.
We understand and agree, but the review is not concluded and the protocol is our guideline until the finish of the review and it is already complete. We continue working on the review, filtering, deepening and refining it. Nevertheless, the number of its register is mentioned in the protocol publication and it can be a good link to direct readers for the review in short time. Instead a link to PROSPERO register will be provided. Furthermore when we finished the review we will take care of your good suggestion and place a link on the methodology to access the full protocol.
Minor comments:
line 163 its a protocol, not a review
Yes, the present paper is the protocol, but designing the umbrella review. In this sense the PictureChart (Figure 1) represents the whole procedure of this running review step by step and the team members and tasks. We apologize if it was not clear and tried to clarify it (marked with different colour).
164-165-I am not sure if this review can provide basis for a new health policy, depends on the results
Yes, perfect! It is so ambitious. Changed in the sense we hope to contribute to new health policies.
232 to exclude-I think that a verb is missing here
Yes. It means “will be excluded” and was changed.
Table 2 number of participants-ais it possible to provide it so exact for animals
Thank you, we clarify that this is just an example including … and a legend.
Round 3
Reviewer 2 Report
I think that the manuscript is nearly ready now.
Please move the sentence "Reliability across researchers should be improved 260 after further meetings to discuss disagreements under this protocol application" to the discussion and discuss there. I suggest that the authors read the manuscript again and correct some small spelling issues.
Author Response
Dear Reviewer,
We deeply thank you for your very constructive suggestions and all the attention to details in order to improve our manuscript. Authors are really grateful to your contribution.
We addressed all you minor suggestions and signal it in red on the manuscript.
Reviewer comments:
I think that the manuscript is nearly ready now.
Please move the sentence "Reliability across researchers should be improved 260 after further meetings to discuss disagreements under this protocol application" to the discussion and discuss there.
Done. Thank you, it fits better on the discussion for sure.
I suggest that the authors read the manuscript again and correct some small spelling issues.
We have made automatic Grammarly correction and 2 authors manual revision of spelling. We believe that now it is acceptable.
